# Modeling Worlds in Text

**Prithviraj Ammanabrolu**
School of Interactive Computing
Georgia Institute of Technology
raj.ammanabrolu@gatech.edu

**Mark O. Riedl**
School of Interactive Computing
Georgia Institute of Technology
riedl@cc.gatech.edu

## Abstract

We provide a dataset that enables the creation of learning agents that can build knowledge graph-based world models of interactive narratives.[1] Interactive narratives—or text-adventure games—are partially observable environments structured as long puzzles or quests in which an agent perceives and interacts with the world purely through textual natural language. Each individual game typically contains hundreds of locations, characters, and objects—each with their own unique descriptions—providing an opportunity to study the problem of giving language-based agents the structured memory necessary to operate in such worlds. Our dataset provides 24198 mappings between rich natural language observations and: (1) knowledge graphs that reflect the world state in the form of a map; (2) natural language actions that are guaranteed to cause a change in that particular world state. The training data is collected across 27 games in multiple genres and contains a further 7836 heldout instances over 9 additional games in the test set. We further provide baseline models using rules-based, question-answering, and sequence learning approaches in addition to an analysis of the data and corresponding learning tasks.

## 1   Introduction

We seek to create agents that exhibit human-like capabilities such as commonsense reasoning and natural language understanding in interactive and situated settings. Interactive narrative environments provide a critical stepping stone in this pursuit towards creating learning agents that can produce contextually relevant and goal-driven natural language [Côté et al., 2018, Urbanek et al., 2019, Hausknecht et al., 2020]. They require agents to observe textual descriptions and then act upon the world using natural language with the aim of completing a long term goal or quest as seen in Figures 1, 2. We focus on two of the core challenges faced by learning agents in these environments—as identified in prior work—knowledge representation and a combinatorially sized state-action space.

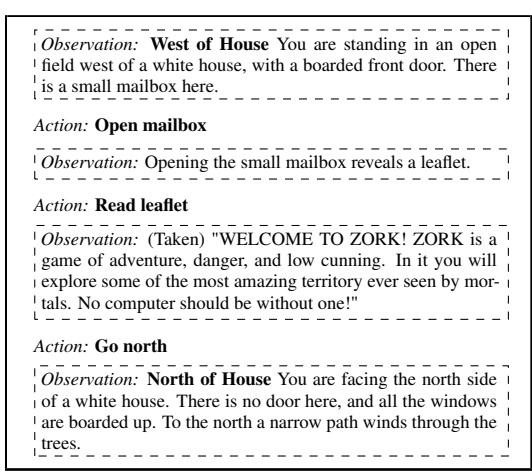

*Observation:* **West of House** You are standing in an open field west of a white house, with a boarded front door. There is a small mailbox here.

*Action:* **Open mailbox**

*Observation:* Opening the small mailbox reveals a leaflet.

*Action:* **Read leaflet**

*Observation:* (Taken) "WELCOME TO ZORK! ZORK is a game of adventure, danger, and low cunning. In it you will explore some of the most amazing territory ever seen by mortals. No computer should be without one!"

*Action:* **Go north**

*Observation:* **North of House** You are facing the north side of a white house. There is no door here, and all the windows are boarded up. To the north a narrow path winds through the trees.

Figure 1: Excerpt from *Zork1*.

The **knowledge representation** challenge rises from the fact that interactive narratives span many distinct locations, each with unique descriptions,

---

[1] Dataset can be found here https://github.com/JerichoWorld/JerichoWorld

35th Conference on Neural Information Processing Systems (NeurIPS 2021), Sydney, Australia.

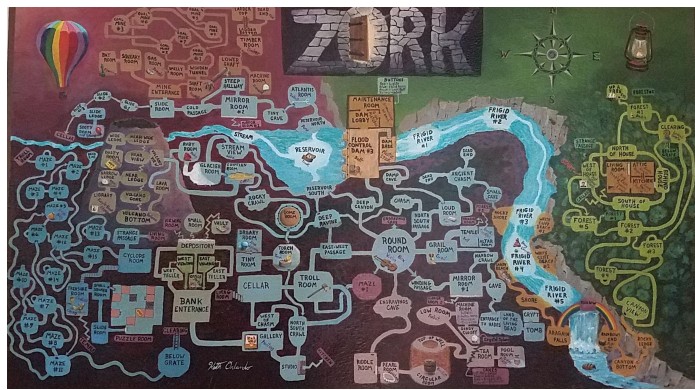

Figure 2: A map showcasing the size and complexity of the world of *Zork* by artist *ion_bond*.

objects, and characters as seen can be seen in Figure 2. Players move by issuing navigational commands, which can convey Euclidean space like *go West* or non-Euclidean span like *step into portal*, warping the agent to an entirely new section of the world. To cope with such challenges, humans often create structured memory aids such as hand drawn maps when attempting to play these games. A good knowledge representation can assist with long-term action dependencies that often arise in game quests (as well as real world environments). An example of a long-term dependency is a key being found in one location that opens a lock on a chest in an entirely different section of the map. For an agent to learn this relationship, it must be able to replicate the sequence of picking up the key and unlocking the chest while not being distracted by interstitial actions and states.

Long-term action dependencies are made challenging by two aspects of interactive narrative environments, which are also present in real-world environments. First, these environments are **partially observable** in the sense that an agent only has local observability. Second, interactive narrative environments have a **combinatorially-sized natural language state-action space**. For example, in the cannonical game *Zork1* an action can consist of up to five-words from a relatively modest vocabulary of 697 words, resulting in $\mathcal{O}(697^5) = 1.64 \times 10^{14}$ possible actions at every step—though the number of *valid actions* that are gramatically coherent and contextually relevant is significantly smaller. This makes exploration sample-inefficient, making it harder to learn the relationship between actions that are temporally distant from each other.

The knowledge representation challenges inherent to interactive narrative games give rise to the **Textual-SLAM** problem, a textual variant of Simultaneous Localization And Mapping (SLAM) [Thrun et al., 2005] problem of constructing a map by *inferring information* from one's surroundings while navigating a novel environment. As in humans, the creation of such world models or memory aids in agents—in the form of knowledge graphs—has been shown to be critical in helping automated learning agents operate in these textual worlds [Ammanabrolu and Riedl, 2019, Murugesan et al., 2020, Adhikari et al., 2020, Ammanabrolu and Hausknecht, 2020].

Despite the success of knowledge graphs in addressing these problems, a broad dataset across a diverse set of games mapping text game observations to knowledge graphs does not exist—hindering progress in building of world modeling agents with structured memory. Building off the popular text game simulator Jericho [Hausknecht et al., 2020], we have constructed a dataset dubbed JerichoWorld that maps text game state observations to both the underlying ground truth knowledge graph representations of the game and the set of contextually relevant actions that can be performed in that state. Using this data, we seek to enable development of agents that focus on answering the questions of "What actions make sense for me to perform right now?" and "What have I already done and how will the world change now if I perform a particular action?"—questions relating to the problems natural language understanding, commonsense reasoning, and structured memory. The training set contains 24198 instances across 27 games and the heldout test set contains 7836 instances from 9 games. We further formally define two initial tasks for this dataset focusing on the questions mentioned: (1) Given a textual observation, predict the underlying knowledge graph of the world. (2) Given a textual observation, predict the set of actions that are contextually relevant. Results for three baselines—using rules-based, question-answering, and sequence-learning approaches—are provided in addition to an analysis of the dataset and results themselves.

## 2 Related Work

We constrain our related work section to three primary areas: current interactive narrative benchmarks, world modeling and model-based reinforcement learning, and the use of knowledge graphs in text games. Currently, three primary open-source platforms and baseline benchmarks have been developed so far to help measure progress in this field: *Jericho* [Hausknecht et al., 2020][2] a learning environment for human-made interactive narrative games; *TextWorld* [Côté et al., 2018][3] a framework for procedural generation in text-games; and *LIGHT* [Urbanek et al., 2019][4] a large-scale crowdsourced multi-user text-game for studying situated dialogue. Further extensions and adaptation to some of these benchmarks have been proposed for use in neighboring domains such as vision-and-language navigation [Shridhar et al., 2021], commonsense reasoning [Murugesan et al., 2021], and procedural text understanding [Tamari et al., 2021]. Our work builds on the Jericho environment.

Work on world models in learning agents have recently been inspired by theories of how humans form mental models of the world [Jancke, 2000, Ha and Schmidhuber, 2018]. When in the form of predictive probabilistic generative models of the world, they can be used in model-based reinforcement learning tasks [Sutton and Barto, 1998, Arulkumaran et al., 2017, Schrittwieser et al., 2019]. In such cases, a learning agent attempts to learn the underlying environment dynamics at the same time as a policy, often using information about one to inform the other. Ha and Schmidhuber [2018] take this one step further by replacing a environment entirely with the agent's own learned world model and training a control policy there. All of these methods have been shown to have the added benefits of significantly improving sample efficiency as the agent is now able to (at least partially) simulate the environment via the world model.

In all of the world modeling cases mentioned, the state representations that the models are conditioned on are drawn directly from the existing base environments, e.g. raw pixel game screens in the case of the Arcade Learning Environment [Bellemare et al., 2013] or other visual games such as Sokoban [Bamford and Lucas, 2020]. In the case of human-made text games, however, knowledge graphs—not directly provided by existing text game learning frameworks—have been shown to be superior state representations when compared to just the textual observations by themselves. They aid in the challenges of partial observability/knowledge representation [Ammanabrolu and Riedl, 2019, Adhikari et al., 2020, Sautier et al., 2020], combinatorial state-action spaces [Ammanabrolu and Hausknecht, 2020, Ammanabrolu et al., 2020b], and commonsense reasoning [Ammanabrolu and Riedl, 2019, Murugesan et al., 2020, 2021, Dambekodi et al., 2020].

Closest in spirit to this work is the Jericho-QA dataset [Ammanabrolu et al., 2020b], a question-answering dataset tuned to text games that enables agents to identify common objects in the world and their attributes. It does not have information regarding the full underlying knowledge graph state or valid actions, however. As far as we know, ground truth knowledge graph state representation dataset across a diverse set of human-made text games is not currently available in any of the primary text game benchmarks mentioned previously, hindering the ability to create agents with structured memory in the form of graph-based world models.

## 3 JerichoWorld

Côté et al. [2018] and Hausknecht et al. [2020] define text games as Partially-Observable Markov Decision Processes. A game can be represented as a 7-tuple of $\langle S, T, A, \Omega, O, R, \gamma \rangle$ representing the set of environment states, *mostly deterministic conditional transition probabilities between states*, the vocabulary or words used to compose text commands, observations returned by the game, observation conditional probabilities, reward function, and the discount factor respectively. Drawing from this definition, each instance of our dataset takes the tuples of $\langle s_t, a_t, s_{t+1}, r_{t+1} \rangle$ where $s_t$ and $s_{t+1}$ are two subsequent states with $a_t$ being the action used to transition states and $r_{t+1}$ is the observed reward for some step $t$.

To collect the $\langle s_t, a_t, s_{t+1}, r_{t+1} \rangle$ tuples we implement a basic agent that explores the game along a trajectory corresponding to a *game walkthrough*. Game walkthroughs are texts describing the solutions to games, generally retrieved from the internet, but already part of the Jericho framework.

---

[2]https://github.com/microsoft/jericho
[3]https://github.com/microsoft/textworld
[4]https://parl.ai/projects/light

Walkthroughs, however, only present one possible solution to a game and solve all the core puzzles required to complete a game with the maximum possible score. To achieve greater coverage of the game's state space, our data collection agent stops off to explore by executing random valid actions for $n$ steps before resetting to the walkthrough. One such collected state—a part of the full tuple mentioned—is detailed below.

The textual observations consist of descriptions of the location and inventory as well as the game engine response to the previous action performed. For example:

```
Game: ztuu
Location: Cultural Complex This imposing ante-room, the center of what was apparently the cultural center
    of the GUE, is adorned in the ghastly style of the GUE's "Grotesque Period." With leering gargoyles,
    cartoonish friezes depicting long-forgotten scenes of GUE history, and primitive statuary of pointy-
    headed personages unknown (perhaps very, very distant progenitors of the Flatheads), the place would
    have been best left undiscovered. North of here, a large hallway passes under the roughly hewn
    inscription "Convention Center." To the east, under a fifty-story triumphal arch, a passageway the
    size of a large city boulevard opens into the Royal Theater. A relatively small and unobtrusive sign
    (perhaps ten feet high) stands nearby. South, a smaller and more dignified (i.e. post-Dimwit) path
    leads into what is billed as the "Hall of Science." You can see a pair of razor-like gloves here.
Observation: You put on the razor-like gloves.
Inventory:
    You are carrying:
    a brass lantern (providing light)
    a pair of glasses
    four candy bars:
        a ZM$100000
        a Multi-Implementeers
        a Forever Gores
        a Baby Rune
    a cheaply-made sword
Prev Act: put on gloves
```

We further provide the set of objects that are found in both the agent's inventory and surroundings, including textual descriptions for each of the objects. Attributes for each of these objects are also included are acquired by decompiling the games, following [Ammanabrolu et al., 2020b]. For example:

```
Inventory Objects:
    candy: Which do you mean, the ZM$100000, the Multi Implementeers, the Forever Gores or the Baby Rune?
    Implementeers: The profiles on the wrapper of this delicacy look more like Moe, Larry, and Curly than
        those of your favorite Implementeers (presumably, Marc, Mike, and David.)
    Forever/Gores: The wrapper of this bar pictures the Milky Way, but the stars are all blood red. Kids
        love them.
    sword: This is a cheaply made sword of no antiquity whatsoever. With regard to grues or other
        underworldly denizens, your weapon is as likely to engender laughter as fear.
    rune: The label is covered with mystical runes, the meanings of which elude you.
    glasses: The owner of these glasses had an indeterminate vision problem, because the lenses have both
        been crushed underfoot. The vision problem, of course, has been solved.
    lantern: The lantern, while of the cheapest construction, appears functional enough for the moment.
        Your best hope is that it stays that way. It looks like the lamp has gone through a few cycles of
        impact revitalization.
Inventory Attributes:
    glasses: clothing
    gloves: clothing
    sword: animate, equip
    lantern: animate, equip
Surrounding Objects:
    gargoyles: Unless you are inordinately masochistic, the less time spent examining the artwork, the
        better.
    east: You see nothing special about the east wall.
    tunnel: The tunnel leads west.
    gloves: The razor like gloves would be very attractive for an axe murderer. And they're just your size.
    south: You see nothing special about the south wall.
    sign: The sign indicates today's performance, which (in honor of the festivities in the Convention
        Center) is "A Massacre on 34th Street."
Surrounding Attributes:
    gloves: clothing
    tunnel: animate
    sign: animate
```

We further provide the ground truth knowledge graph representing the world state corresponding to these textual observations. The ground truth knowledge graph is a set of tuples $\langle s, r, o \rangle$ such that $s$ is a subject, $r$ is a relation, and $o$ is an object. It reflects information on the current state such as objects and attributes and is extracted from the game engine by traversing the engine's internal representation and converting it to human readable form. Relations are defined on the basis of traversal operations

in the game engine's internal representation, e.g. "in" and "have" signify parent-child ownership for locations and inventory respectively. Tuples in the graph that cannot be directly extracted from the observation text are marked so in the dataset. For example:

```
Graph: [sign, in, Cultural Complex], [you, have, Forever Gores], [you, have, ZM$100000], [you, have, Baby
    Rune], [tunnel, in, Cultural Complex], [you, in, Cultural Complex], [you, have, brass lantern], [you,
    have, glasses], [decoration, in, Cultural Complex], [you, have, cheaply-made sword], [you, have,
    Multi-Implementeers], [you, have, razor-like gloves], [glasses, is, clothing], [gloves, is, clothing],
    [sword, is, animate], [tunnel, is, animate], [sign, is, animate], [lantern, is, animate], [sword, is,
    equip], [lantern, is, equip]
```

Valid actions are defined by Hausknecht et al. [2020] as the set of actions guaranteed to cause a change in the current world state and are identified by the Jericho framework. For example in one particular state me might have the following valid actions:

```
Valid Actions: west, turn lantern off, east, south, put multi down, put forever down, put lantern down,
    put rune down, put glasses down, put sword down, take razor off, put on glasses, examine glasses,
    lower razor, throw multi, throw lantern, put multi in glasses, north
```

## 3.1 Dataset Analysis

| Game | No. Samples | Input Vocab Size | Avg. Obs Token Len. | Avg. Graph Triple Len. | Avg. No. Valid Actions | Avg. Surround. Objects |
|---|---|---|---|---|---|---|
| **Training games** | | | | | | |
| wishbringer | 560 | 1043 | 136.54 | 4.00 | 10.35 | 8.51 |
| snacktime | 168 | 468 | 190.08 | 2.33 | 4.82 | 5.52 |
| tryst205 | 1052 | 871 | 136.24 | 7.81 | 14.30 | 8.38 |
| enter | 440 | 470 | 219.06 | 14.79 | 18.04 | 9.23 |
| omniquest | 784 | 460 | 79.96 | 8.02 | 21.50 | 5.30 |
| zork3 | 1142 | 564 | 137.68 | 6.59 | 12.72 | 5.26 |
| zork2 | 584 | 684 | 154.90 | 7.82 | 29.66 | 5.73 |
| inhumane | 1004 | 409 | 90.24 | 3.86 | 4.31 | 2.48 |
| 905 | 504 | 296 | 100.91 | 11.69 | 13.60 | 12.24 |
| loose | 16 | 1141 | 140.38 | 10.12 | 2.12 | 9.00 |
| murdac | 1914 | 251 | 80.76 | 4.30 | 8.67 | 1.63 |
| moonlit | 684 | 669 | 131.62 | 12.10 | 9.20 | 11.61 |
| dragon | 894 | 1049 | 182.79 | 11.64 | 13.13 | 12.29 |
| jewel | 1418 | 657 | 119.08 | 7.21 | 13.82 | 5.15 |
| weapon | 294 | 481 | 230.41 | 29.79 | 9.65 | 35.68 |
| karn | 2196 | 615 | 138.87 | 13.24 | 26.36 | 8.44 |
| zenon | 402 | 401 | 101.52 | 5.01 | 5.97 | 3.95 |
| acorncourt | 474 | 343 | 323.38 | 36.14 | 20.18 | 16.08 |
| ballyhoo | 2132 | 962 | 127.08 | 7.25 | 15.39 | 7.11 |
| yomomma | 884 | 619 | 129.06 | 3.00 | 16.11 | 5.52 |
| enchanter | 1714 | 722 | 133.56 | 14.83 | 45.27 | 7.40 |
| gold | 2082 | 728 | 166.96 | 15.76 | 25.03 | 12.92 |
| huntdark | 344 | 539 | 162.33 | 13.01 | 6.33 | 6.90 |
| afflicted | 574 | 762 | 165.13 | 2.91 | 17.34 | 11.85 |
| adventureland | 870 | 398 | 87.41 | 6.99 | 9.02 | 5.17 |
| reverb | 722 | 526 | 101.92 | 5.23 | 9.04 | 4.78 |
| night | 346 | 462 | 49.92 | 10.17 | 4.55 | 3.37 |
| **overall train** | 24198 | 11056 | 133.30 | 9.74 | 17.41 | 7.70 |
| **Testing games** | | | | | | |
| deephome | 630 | 760 | 147.33 | 10.20 | 15.31 | 7.15 |
| balances | 990 | 452 | 107.15 | 7.61 | 13.04 | 3.85 |
| ludicorp | 2210 | 503 | 88.32 | 9.47 | 9.27 | 4.60 |
| pentari | 276 | 472 | 130.34 | 3.46 | 3.72 | 2.84 |
| detective | 434 | 344 | 105.97 | 2.80 | 5.72 | 2.16 |
| ztuu | 462 | 607 | 170.89 | 11.97 | 18.39 | 7.94 |
| zork1 | 886 | 697 | 109.70 | 6.46 | 13.02 | 4.54 |
| library | 654 | 510 | 154.40 | 9.18 | 4.59 | 10.20 |
| temple | 1294 | 622 | 138.07 | 10.77 | 8.56 | 8.78 |
| **overall test** | 7836 | 11056 | 118.92 | 8.71 | 10.30 | 5.86 |

Table 1: Dataset statistics across the games. All games together have a combined input vocabulary of size 11056. There are 17 unique graph relations and 6985 unique graph entity names (i.e. locations, characters, and objects) across all the games. Vocabulary files are provided in the dataset.

Table 1 presents statistics for our data in the form of showing vocab sizes, and average lengths of different data fields. The dataset as a whole has an input vocabulary size of 11056—this is the superset of the vocabulary that can be used to act in any of these games. It is worth noting, that the output vocabulary size—determined by the observations—is not restricted, whereas the input vocabulary is restricted by the game developer. As seen later when we introduce models for these

tasks, this means that subword based tokenization [Kudo, 2018] for processing inputs is the most effective way of avoiding unknown tokens.

The training and testing games both cover a wide range of genres as noted by Hausknecht et al. [2020], Ammanabrolu et al. [2020b]—e.g. *905* is a everyday slice-of-life simulator in which a character walks around a house preparing for work, *afflicted* is a monster horror game, *ballyhoo, detective* are murder mysteries, and *karn, zork1* are traditional fantasies. On average, the observation token, graph, and valid action lengths are comparable across both the training and testing games. Outliers in these metrics usually represent game-specific challenges. For example, *acorncourt* has the highest observation token and graph length counts by far. This is because the game is focused heavily on object collection and so contains more entities on average than others. In a similar vein, *enchanter* has significantly more valid actions than other games. This is due to the game being focused on constantly discovering valid actions in the form of spells and their effects by casting them—everything from healing yourself to causing an object to give off light. It is worth noting that many of these spells appear and have similar effects in other fantasy text games. These are some examples of challenges that players must overcome to be successful in these worlds.

## 4 Benchmarks

This section introduces the two primary tasks using JerichoWorld required for world modeling in learning agents: *knowledge graph prediction* and *valid action prediction*. We then introduce baseline models for each of the tasks, report zero-shot results on the testing games, and analyze performance.

### 4.1 Knowledge Graph Prediction

The first world modeling task involves predicting a knowledge graph from the current set of textual observations. Recall that our dataset takes the form of tuples of $\langle s_t, a_t, s_{t+1}, r_{t+1} \rangle$ where $s_t$ and $s_{t+1}$ are two subsequent states with $a_t$ being the action used to transition states and $r_{t+1}$ is the observed reward. This task is to predict $s_{t+1}^{\text{graph}}$, a set of knowledge graph relations, given the textual observations $s_t^{\text{obs}}$, the previous state's graph $s_t^{\text{graph}}$, and action $a_t$ for all samples in the dataset. We present three baseline models for this task.

**Rules.** Following Ammanabrolu and Hausknecht [2020], [5] we extract graph information from the observation using information extraction tools such as OpenIE [Angeli et al., 2015] in addition to some hand-authored rules to account for the irregularities of text games.

**Question-Answering.** This baseline comes from the Q*BERT agent described in Ammanabrolu et al. [2020b]. [6] It is trained on the SQuAD 2.0 [Rajpurkar et al., 2018], the Jericho-QA text game question answering dataset [Ammanabrolu et al., 2020b] on the same set of training games as found in JerichoWorld, and then on JerichoWorld itself by formatting our dataset in the style of questions and answers when possible. It uses the ALBERT [Lan et al., 2020] variant of the BERT [Devlin et al., 2018] natural language transformer to answer questions and populate the knowledge graph via a few hand-authored rules from the answers. Examples of questions asked include: "What is my current location?", "What objects are around me?".

**Seq2Seq.** We further introduce a encoder-decoder based sequence-to-sequence learning approach [Sutskever et al., 2014] inspired by the transformer model BART [Lewis et al., 2020]. The model architecture is shown in Figure 3 and consists of a bidirectional encoder such as BERT [Devlin et al., 2018] that takes the full set of textual observations—including location and inventory descriptions—as input and an autoregressive decoder such as GPT-2 [Radford et al., 2019] which takes in the current graph and learns to predict the graph sequence shifted by a token. The weights of the encoder are fine-tuned from BERT's original weights on both the graphs, in triple form, and the textual observations taken from the training games using a masked language modeling loss. The decoder is not pre-trained. During test time, only the starting token is given to the decoder and it decodes the graph token by token via beam search until an end-of-sequence token is reached.

**Metrics.** For this task, we report two types of metrics (Exact Match or EM and F1) operating on two different levels—at a *graph* tuple level and another at a *token* level. EM checks for accuracy or direct

---

[5] https://github.com/rajammanabrolu/KG-A2C
[6] https://github.com/rajammanabrolu/Q-BERT

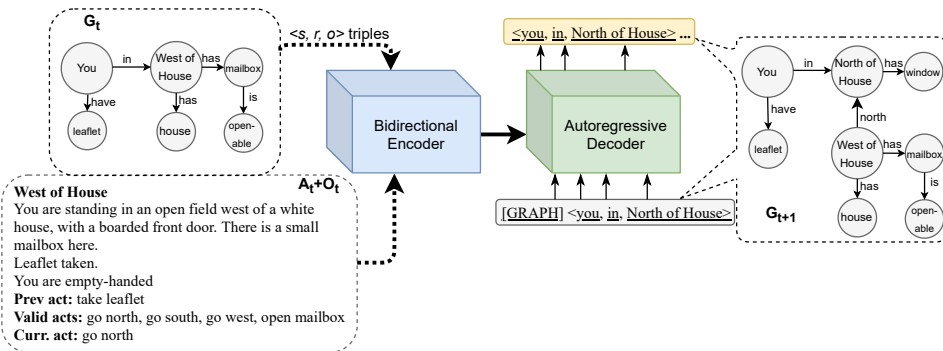

Figure 3: A description of the Seq2Seq architecture for knowledge graph prediction task with a bidirectional encoder and autoregressive decoder. A similar architecture is used for the Seq2Seq model shown in the valid action prediction task.

overlap between the predictions and ground truth, while F1 is a harmonic mean of predicted precision and recall. The graph level metrics are based on matching the set of ⟨*subject, relation, object*⟩ triples within the graph, all three tokens in a particular triple must match a triple within the ground truth graph to count as a true positive. The token level metrics operate on measuring unigram overlap in the graphs, any relations or entities in the predicted tokens that match the ground truth count towards a true positive.

**Analysis.** Table 2 presents a breakdown of the results for this task across the testing games on all the baseline models presented. There are a few main trends to note in these results. The first is that the question-answering (QA) approach significantly outperforms both the Rules and Seq2Seq approaches on average across all the testing games. The QA method used is extractive. This means that the system is trained to pick out answers by highlighting spans in the input context that best answers a question. The Rules approach also functions similarly but is not trained in any way on our data. This is inherently a simpler problem formulation than the Seq2Seq approach—which seeks to generate the graph by decoding token by token—but has its limitations.

These limitations are seen in the relative differences between the magnitudes of the graph and token metrics for these approaches. Both QA and Rules have significantly lower graph metrics than token metrics, a phenomenon not observed in the Seq2Seq model. In other words, the right information is extracted but is potentially not well shaped into knowledge graph form. We hypothesize that this implies two things. (1) That these systems likely *over-extract* by extracting more information than is strictly necessary. Take for example the sample observation seen in Figure 3: *"You are standing in an open field west of a white house, with a boarded front door."*. QA when asked the question "What is my location?" answers: "open field west of a white house, with a boarded front door". Seq2Seq, in contrast, is trained to map this sentence more tersely to: ⟨*you, in, West of House*⟩. (2) Both QA and Rules use hand-crafted rules to put the graphs together once information has been extracted either through the core QA model or OpenIE. We see here that while over-extraction can be beneficial for the token metrics—it makes it difficult to create a set of graph construction rules that generalize well across games with different structures, resulting in relatively lower graph metrics.

On the other hand, the main advantage of the Seq2Seq approach is that it is not extractive and trained directly on the graphs found in the dataset. This means that it is potentially able to infer facts that are not directly present in the input context. Recall that text games are *partially observable* and so the textual observations themselves may potentially be incomplete. An example of such an observation is: *"You see a locked chest in front of you in the cellar."*. The ground truth graph for this would be: ⟨*you, in, Cellar*⟩, ⟨*chest, in, cellar*⟩, ⟨*chest, is, lockable*⟩, ⟨*sword, in, chest*⟩. The last fact in the graph, the sword being in the chest, is not revealed to you via the observation until you open the chest and thus cannot be predicted by extractive approaches like Rules and QA. This gives models like Seq2Seq—that are trained directly on the graph—the ability to perform commonsense inference by potentially filling in information missing from the partially observable text inputs. It further implies that extractive models, in their current form, will not be able to achieve perfect performance.

The main limitation of the Seq2Seq model, however, is that this non-extractive framing—given that every token is decoded autoregressively, requiring a prediction at every step over the entire combined

| Expt. | | Rules | | | | Question-Answering | | | | Seq2Seq | | | |
| Metric | | Graph | | Token | | Graph | | Token | | Graph | | Token | |
| Game | Size | EM | F1 | EM | F1 | EM | F1 | EM | F1 | EM | F1 | EM | F1 |
|---|---|---|---|---|---|---|---|---|---|---|---|---|---|
| zork1 | 886 | 3.72 | 4.46 | 6.08 | 8.42 | 24.56 | 24.88 | 43.93 | 48.31 | 12.44 | 12.96 | 18.01 | 21.12 |
| library | 654 | 7.61 | 12.87 | 10.33 | 26.74 | 29.14 | 31.46 | 49.78 | 52.76 | 18.42 | 18.89 | 20.26 | 20.84 |
| detective | 434 | 1.39 | 4.55 | 7.51 | 10.23 | 34.45 | 36.23 | 60.28 | 63.21 | 26.86 | 29.48 | 35.86 | 35.86 |
| balances | 990 | 9.17 | 11.9 | 32.53 | 36.09 | 41.22 | 41.85 | 85.81 | 86.18 | 8.19 | 9.04 | 17.6 | 18.86 |
| pentari | 276 | 6.44 | 10.22 | 16.48 | 23.36 | 28.96 | 30.12 | 65.02 | 69.54 | 22.18 | 23.54 | 25.48 | 27.72 |
| ztuu | 462 | 4.94 | 10.06 | 14.4 | 21.74 | 22.17 | 26.26 | 49.44 | 49.82 | 16.89 | 16.89 | 17.19 | 17.87 |
| ludicorp | 2210 | 5.1 | 8.37 | 14.47 | 18.48 | 41.44 | 46.74 | 57.58 | 60.95 | 12.94 | 14.18 | 14.8 | 15.42 |
| deephome | 630 | 0.49 | 0.64 | 3.34 | 3.86 | 4.42 | 4.66 | 9.31 | 9.84 | 8.38 | 10.47 | 13.25 | 13.25 |
| temple | 1294 | 2.48 | 3.36 | 7.42 | 9.44 | 36.84 | 39.86 | 48.98 | 49.17 | 16.48 | 18.52 | 22.48 | 24.34 |
| **overall** | 7836 | 4.70 | 7.25 | 13.08 | 17.50 | **32.78** | **35.48** | **53.58** | **55.74** | 14.29 | 15.54 | 18.80 | 19.96 |

Table 2: Results for the Knowledge Graph Prediction task. Overall indicates a size weighted average. All experiments are evaluated over three random seeds with standard deviations not exceeding $\pm 2.8$ in any overall category.

entity and relation vocabulary length of 7002—is a significantly more difficult problem than the other approaches. It is likely that that such non-extractive approaches will have to simplify the problem by adding constraints that account for properties of knowledge graphs (e.g. graph are sets of tuples and the same tuple in a set cannot be decoded twice).

## 4.2 Valid Action Prediction

The second world modeling task involves predicting the set of valid actions from the current set of textual observations. Given the data $\langle s_{t-1}, a_{t-1}, s_t, r_t \rangle$ (note the change in indexing), this task is formally defined as: predict the set of valid actions for the subsequent state $s_t^{\text{valid}}$ given the current state text observation $s_t^{\text{obs}}$, current knowledge graph $s_t^{\text{graph}}$, previous valid actions $s_{t-1}^{\text{valid}}$, and action $a_{t-1}$ that caused the state change for all individual samples across the dataset. This task requires linguistic priors in the form of commonsense reasoning and a knowledge of affordances—e.g. *open mailbox* is a more reasonable action to take in most situations than *eat mailbox*.

We present a single baseline for this task. We developed a **Seq2Seq** model that is identical to that presented for the Knowledge Graph Prediction task, except adapted to Valid Action Prediction. That is, it performs sequence learning on the valid actions token by token. Extractive approaches like QA are not possible for valid action prediction given that the verbs in the action—e.g. *take, get, put, swing, go*—are not often found anywhere within the observation. The Seq2Seq approach thus decodes actions token by token from the entire combined output vocab of 11056 (see Table 1) at every step until a special end-of-sequence tag is reached.

**Metrics.** For this task, we adapt the graph level Exact Match (EM) and F1 metrics as described in the previous task to actions. In other words, positive EM or F1 happens only when all tokens in a predicted valid action match one in the gold standard set. Given that most valid actions have less than four tokens, we do not use standard Seq2Seq metrics—such as BLEU [Papineni et al., 2002]—intended for measuring $n$-gram overlap in longer sequences. We do not report token unigram overlap, as with the knowledge graph task as here, because predicted actions are required to match gold standard actions exactly in order to be executable in the game.

| Game | Size | EM | F1 |
|---|---|---|---|
| zork1 | 886 | 16.65 | 17.85 |
| library | 654 | 15.13 | 16.88 |
| detective | 434 | 18.19 | 21.12 |
| balances | 990 | 16.19 | 18.23 |
| pentari | 276 | 23.39 | 25.87 |
| ztuu | 462 | 14.75 | 15.13 |
| ludicorp | 2210 | 20.1 | 20.86 |
| deephome | 630 | 14.71 | 14.86 |
| temple | 1294 | 20.34 | 22.14 |
| **overall** | 7836 | 18.10 | 19.44 |

Table 3: Results for the Valid Action Prediction task. Overall indicates a size weighted average. All experiments are evaluated over three random seeds with standard deviations not exceeding $\pm 3.7$ in any overall category.

**Analysis.** Table 3 shows the results for the valid action prediction task on all the testing games for the Seq2Seq baseline. Recall that an EM of 20 means that if there were 100 gold standard valid actions in an instance, the model predicted 20 of them exactly. Based on this, we further note a trend in Table 3 that there is a negative correlation between error rates and the average number of valid actions as seen in Table 1. That is, the more the average number of gold standard valid actions per instance in a game, the more predicted actions match. Games like *ztuu, deephome, balances* have

a high number of gold standard average valid actions and lower performance than games like *pentari, ludicorp, detective, temple* which have a low number of average valid actions. The expected result would be that a model is able to learn a smaller sequence more effectively than a larger one—implying that a smaller number of gold standard valid actions per instance would lead to more matches. We hypothesize that this is likely due to the fact that the model best learns common actions found across all games first before learning potentially more fine grained actions—effectively a label imbalance issue across the valid actions in the dataset. E.g. navigation actions like *go north* are found much more often than actions like *hit monster with sword*—which are usually found in only a handful of fantasy games. When performing zero-shot prediction on testing games, the model thus predicts these common actions with higher confidence than the more fine grained ones. Testing games with a smaller number of average gold standard valid actions also tend to have a larger proportion of uncommon actions—thus posing more of a challenge for the Seq2Seq model.

## 5   Conclusions and Future Work

This paper presents the JerichoWorld dataset and corresponding benchmarks that seek to drive progress in textual world modeling. This primarily involves two key challenges behind the creation of agents that can understand and generate natural language in a diverse set of interactive and situated settings such as text games. Our dataset provides mappings from textual observations to ground truth knowledge graph states to enable agents to learn to infer the state of the world—alleviating the *knowledge representation* or *Textual-SLAM* challenge. A key insight from a comparison of baseline models shows that a promising future direction lies in *inferring* the knowledge graph world state through commonsense reasoning rather than *extracting* this information due to partial observability.

A second world modeling task revolves around tacking the *combinatorially-sized action space* of text games. The dataset also provides mappings from textual observations to valid actions—i.e. the set of contextually relevant actions guaranteed to change the world in any state. A qualitative analysis of a state-of-the-art Seq2Seq model adapted to the domain and trained for this task suggests that while learning to conditionally generate commonly occurring actions across a large set of games might be relatively easy, learning to generate specific and contextually relevant actions provides a significantly more difficult challenge. Current performance by state-of-the-art models across both these tasks suggests that there is much space for improvement though recent works such as  Fan et al. [2019a], Ive et al. [2019], Schmitt et al. [2020] might provide useful starting points.

There are many more tasks that can be framed for other challenges related to world modeling from this dataset. Some immediate examples: (1) offline reinforcement learning for game agents through imitation learning—predicting the sequence of actions that finish the game based on walkthroughs and reward information; (2) knowledge graph verbalization, a form of the standard data-to-text natural language processing task [Wiseman et al., 2017], in which we learn to generate text that is conditioned on a knowledge graph; and (3) description generation conditioned on the names of various objects, locations, and characters—with applications in long-form text generation domains such as automated storytelling [Martin et al., 2018, Fan et al., 2019b] and procedural generation of interactive narratives [Ammanabrolu et al., 2020a, Walton et al., 2020].

## 6   Broader Impacts

Text games are simplified analogues for systems capable of long-term dialogue with humans, such as in assistance with planning complex tasks, and also discrete planning domains such as logistics. Our focus is on helping agents to better model such worlds, enabling greater efficiency for agents training to produce such contextually relevant language.

The data is collected from games containing situations of non-normative language usage—describing situations that fictional characters may engage in that are potentially inappropriate, and on occasion impossible, for the real world such as running a troll through with a sword. Instances of such scenarios are mitigated by careful curation of the games that the data is collected from. The original Jericho framework [Hausknecht et al., 2020]—further verified by us in this work—uses a curated set of games found not to contain extreme examples of non-normative language usage. This is based on manual vetting and (existing) crowd-sourced reviews on the popular interactive narrative forum IFDB.[7]

---

[7]https://ifdb.org/

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

# A Appendix

We would first like to note the presence of a complementary dataset of observation-action pairs created by humans on the ClubFloyd online Interactive Narrative forum.[8] This dataset appears in both Ammanabrolu and Hausknecht [2020] and Yao et al. [2020] with the latter using it to tune a GPT-2 model for valid action prediction. To prevent data leakage from human transcripts to our test games, we do *not* use this dataset to pre-train or tune our models.

The rest of this Appendix first provides additional samples for the dataset for qualitative purposes and then provides training details for the baseline models.

## A.1 Dataset

The games used in the Jericho suite and here are all open sourced freeware. The walkthroughs required to create the oracle agents for the collection of data for the games were drawn from various sources on the internet and errors were corrected manually. We provide our data at https://github.com/JerichoWorld/JerichoWorld under an MIT license. We provide 3 samples drawn from different games in the full dataset to help the readers better understand the diversity of text there.

```
Game: 905
Location:
    Bedroom (in bed)
    This bedroom is extremely spare, with dirty laundry scattered haphazardly all over the floor. Cleaner
        clothing can be found in the dresser. A bathroom lies to the south, while a door to the east
        leads to the living room.

    On the end table are a telephone, a wallet and some keys.

    The phone rings.
Observation: You take off the gold watch. The phone rings.
Inventory:
    You are carrying:
      some soiled clothing (being worn)
      a gold watch
Prev Act: take off watch
Inventory Objects:
    gold watch: Apparently it's 9:07. The phone rings.
    soiled clothing: These clothes are a lost cause, sad to say no amount of laundering is going to get
        these stains out.
Inventory Attributes:
    watch: animate, equip
    clothing: animate, equip
Surrounding Objects:
    phone: An ordinary telephone, notable chiefly for being fifteen or twenty years old.
    keys: House keys, car keys, they're all on the same chain.
    end table: A small end table, oak veneer over plywood.
    living room: The living room lies to the east.
    dirty dresser: Just a simple dresser.
    laundry: Shirts, pants, the usual.
    floor, east, south: You see nothing unexpected in that direction. The phone rings.
    wallet: It's a brown leather wallet.
    door: Just a regular door.
Surrounding Attributes:
    keys: animate, equip
    wallet: animate, equip
Graph: [you, have, gold watch], [you, in, bed], [you, have, soiled clothing]
Valid Actions: take phone, get up, take off clothing, take off watch, take keys, close door, take wallet,
    close door, put clothing down, put watch down, put clothing on table, open wallet, put watch down,
    put clothing on phone, put watch on table, put gold on phone, look under bed

Game: deephome
Location:
    Secret Entrance
    This is a rather dark and small room, having only two exits, back north the way you came, from the
        ancestral homes of Tana, or through the heavily barred wooden door before you that leads
        southwest and inward to the abandoned Deephome, abode of the Dwarves in Telleen. It has been
        three hundred years since your people lived here.

    The heavy door stands open, admitting you into Deephome.
Observation: As you touch the finely etched symbol, you hear a click and a whir. Then the door swings open
        before you, opening into the abandoned city of Deephome. Your score has just gone up by five points.
Inventory:
```

---

[8]http://www.allthingsjacq.com/interactive_fiction.html

```
    You are carrying:
      King's Order
      a lantern (providing light)
Prev Act: push mountain
Inventory Objects:
    lantern: This is an old and trusty (not rusty) lantern that has been in your family for centuries. It
        has yet to shut off at an inopportune moment. However, there is a saying in your family..."That
        lantern is bound to go off at an inopportune time sometime!"
    order: The note reads: "Reclaimer: You have the esteemed duty to return to our Mountain Kingdom of
        Deephome and prepare it for our return. There are several things a Reclaimer must do: 1. Restore
        Power to the City 2. Restore Water to the city. 3. Visit each location and make sure it is safe,
        a quick appraisal should be sufficient. 4. Open the City Gates once more. 5. MOST IMPORTANT: Make
         sure the city is SAFE to return to. May the Peace of Kraxis go with you King Derash of the
        Mountain Tana, the year 782 SK."
Inventory Attributes:
    lantern: equip
Surrounding Objects:
    southwest: You see nothing special about the southwest wall.
    house: It is the typical human house, maybe two stories. It is etched into the wood.
    wooden door: This door is made of thick and sturdy wood. It has three symbols on it, a tree, a house,
        and a mountain.
    symbols: On the door there are pictures of a mountain, a tree, and a house.
    tree: The tree symbol looks as if it were etched into the wood.
    mountain: The mountain looks mighty, a high peak among the clouds. It is etched into the wood.
Surrounding Attributes:
    door: unlockable
    symbols: unlock
Graph: [symbols, in, Secret Entrance], [wooden door, in, Secret Entrance], [ground, in, Secret Entrance],
    [you, in, Secret Entrance], [house, in, Secret Entrance], [Kraxis, in, Secret Entrance], [you, have,
    lantern], [mountain, in, Secret Entrance], [you, have, "Kings Order"], [tree, in, Secret Entrance]
Valid Actions: say manaz, push mountain, close wooden, get in southwest, put light down, put order down

Game: reverb
Location:
    Behind the Counter
    You are behind the counter at "Mr. Tasty's Pizza Parlor". To the southwest is the rest of the
        restaurant.

    On the counter is a large pizza box (which is closed).

    You can see a handwritten note here.
Observation: You put the large pizza box on the counter.
Inventory: You are carrying nothing.
Prev Act: push large to counter
Inventory Objects:
Inventory Attributes:
Surrounding Objects:
    southwest: You see nothing special about the southwest wall.
    handwritten note: The note reads: "Stanley, Don't forget to make your delivery to Mr. Calzone, located
        at the San Doppleton Courthouse. You're already on thin ice, kid. One more screwup and you can
        expect to be looking for a new job." The note is signed with the initials "RT". The paper is
        official "Mr. Tasty's" stationery with the name Bob "Tasty" Tasker and lots of balloons and
        smiley faces all over the border. Isn't that cute?
    large pizza box: It's a large, flat, greasy cardboard box. Hastily scrawled on the outside is the word
        "Calzone". Which is weird, because it's clearly a pizza.
    counter: It's a majorly boring counter which you're unfortunately very familiar with.
Surrounding Attributes:
    handwritten note: indoor, readable
    large pizza box: indoor
    counter: indoor
Graph: [metal file, in, large pizza], [you, in, Behind the Counter], [handwritten note, in, Behind the
    Counter], [large pizza, in, large pizza box], [counter, in, Behind the Counter], [large pizza box, in,
    counter]
Valid Actions: get up, take note, take large, examine note, undo large, push note to southwest, push large
    to southwest, push note to counter, push large to counter
```

## A.2   Baselines

The baseline models that are adapted from other works, i.e. the Rules and QA systems, are trained using hyperparameters and methodologies described in their respective works.

### A.2.1   Rules

Following Ammanabrolu and Hausknecht [2020], the exact details regarding knowledge graph updates are found as follows. At every step, given the current state and possible attributes as context. The rest of the triples are extracted using OpenIE [Angeli et al., 2015].

- Linking the current room type (e.g. "Kitchen", "Cellar") to the items found in the room with the relation "has", e.g. $\langle kitchen, has, lamp \rangle$

- All attribute information for each object is linked to the object with the relation "is". e.g. $\langle egg, is, treasure \rangle$

- Linking all inventory objects with relation "have" to the "you" node, e.g. $\langle you, have, sword \rangle$

- Linking rooms with directions based on the action taken to move between the rooms, e.g. $\langle Behind\ House, east\ of, Forest \rangle$ after the action "go east" is taken to go from behind the house to the forest

### A.2.2 Question-Answering

The QA models are trained on the SQuAD 2.0 [Rajpurkar et al., 2018], the Jericho-QA text game question answering dataset on the same set of training games as found in JerichoWorld, and then on JerichoWorld itself by formatting our dataset in the style of questions and answers when possible. Our dataset is formatted in the style of Jericho-QA by templating questions that ask about location, objects (including characters), and attributes. An example of a JerichoWorld dataset example converted to Jericho-QA format is seen below—though we would like to note that this removes much of the information present naturally within our dataset. All other model architecture and hyperparameter details are as seen in Ammanabrolu et al. [2020b].

```
Game: reverb
Location:
    Behind the Counter
    You are behind the counter at "Mr. Tasty's Pizza Parlor". To the southwest is the rest of the
        restaurant.

    On the counter is a large pizza box (which is closed).

    You can see a handwritten note here.
Observation: You put the large pizza box on the counter.
Inventory: You are carrying nothing.
Question: Where am I located? Answer: Behind the Counter
Question: What is here? Answer: large pizza box, handwritten note, southwest
Question: What do I have? Answer: nothing
Question: What attributes does handwritten note have? Answer: indoor, readable
Question: What attributes does southwest have? Answer: indoor
Question: What attributes does large pizza box have? Answer: indoor
```

### A.2.3 Seq2Seq

For both tasks, models were trained until validation accuracy (picked to be a random $10\%$ subset of the training data) did not improve for 5 epochs or 72 wall clock hours on a machine with 4 Nvidia GeForce RTX 2080 GPUs, three times with three random seeds. All models decode using beam search with a beam width of 15 at test time until the end-of-sequence tag is reached. The size of the decoding vocabulary for the action prediction task is 11056 and for the graph prediction task is 6985. Hyperparameters were not tuned and were taken from BART [Lewis et al., 2020].

| Hyperparameter type | Value |
| --- | --- |
| Dictionary Tokenizer | Byte-pair encoding |
| Num. Encoder layers | 6 |
| Num. Decoder layers | 6 |
| Num. encoder and decoder attention heads | 8 |
| Feedforward network hidden size | 4096 |
| Input length | 1024 |
| Embedding size | 768 |
| Batch size | 16 |
| Dropout ratio | 0.1 |
| Gradient clip | 1.0 |
| Optimizer | Adam |
| Learning rate | $1.0 \times 10^{-3}$ |

Table 4: Hyperparameters used to train the Seq2Seq model. It has a total of 232 million trainable parameters.

# B Datasheet

We provide comprehensive documentation of the dataset based on Datasheets for Datasets [Gebru et al., 2018].

## B.1 Motivation

**For what purpose was the dataset created? Was there a specific task in mind? Was there a specific gap that needed to be filled? Please provide a description.** We seek to create agents that exhibit human-like capabilities such as commonsense reasoning and natural language understanding in interactive and situated settings. In pursuit of this goal, we provide a dataset that enables the creation of learning agents that can build knowledge graph-based world models of interactive narratives.

**Who created this dataset (e.g., which team, research group) and on behalf of which entity (e.g., company, institution, organization)?** It was created by Prithviraj Ammanabrolu and Mark Riedl at the Georgia Institute of Technology.

**Who funded the creation of the dataset? If there is an associated grant, please provide the name of the grantor and the grant name and number.** It was funded by the US's Defense Advanced Research Projects Agency (DARPA) as part of a fundamental science research grant Science of Artificial Intelligence and Learning for Open-world Novelty (SAIL-ON https://www.darpa.mil/program/science-of-artificial-intelligence-and-learning-for-open-world-novelty).

## B.2 Composition

**What do the instances that comprise the dataset represent (e.g., documents, photos, people, countries)? Are there multiple types of instances (e.g., movies, users, and ratings; people and interactions between them; nodes and edges)? Please provide a description.** Each instance of our dataset takes the tuples of $\langle s_t, a_t, s_{t+1}, r_{t+1} \rangle$ where $s_t$ and $s_{t+1}$ are two subsequent states of a text game with $a_t$ being the action used to transition states and $r_{t+1}$ is the observed reward for some step $t$. Everything is in text. These are all collected from various text games and examples of instances are found in Appendix A.1.

**How many instances are there in total (of each type, if appropriate)?** The training data has 24198 mappings and is collected across 27 games in multiple genres and contains a further 7836 heldout instances over 9 additional games in the test set.

**Does the dataset contain all possible instances or is it a sample (not necessarily random) of instances from a larger set? If the dataset is a sample, then what is the larger set? Is the sample representative of the larger set (e.g., geographic coverage)? If so, please describe how this representativeness was validated/verified. If it is not representative of the larger set, please describe why not (e.g., to cover a more diverse range of instances, because instances were withheld or unavailable).** The dataset is a sample of the larger set of all possible states in each game. The samples are made to be biased towards states near the walkthroughs required to finish a game.

**What data does each instance consist of? "Raw" data (e.g., unprocessed text or images)or features? In either case, please provide a description.** Data is all in the form of text, either raw or in structured knowledge graph form.

**Is there a label or target associated with each instance? If so, please provide a description.** The data has multiple fields, depending on the tasks defined any of them can be used as labels. E.g. the knowledge graph prediction task has the graph field as the target.

**Is any information missing from individual instances? If so, please provide a description, explaining why this information is missing (e.g., because it was unavailable). This does not include intentionally removed information, but might include, e.g., redacted text** Not all games have human readable attributes for objects—when they do not, these are omitted by leaving the attributes fields blank. All other data is present for all instances.

**Are relationships between individual instances made explicit (e.g., users' movie ratings, social network links)? If so, please describe how these relationships are made explicit.** Instances are grouped together by game through the game field.

**Are there recommended data splits (e.g., training, development/validation, testing)? If so, please provide a description of these splits, explaining the rationale behind them.** We provide a training split of 27 games, and a testing split of 9 games. These are selected on the basis of existing works and each split contains a diverse set of games in terms of genre.

**Are there any errors, sources of noise, or redundancies in the dataset? If so, please provide a description.**

**Is the dataset self-contained, or does it link to or otherwise rely on external resources (e.g., websites, tweets, other datasets)? If it links to or relies on external resources, a) are there guarantees that they will exist, and remain constant, over time; b) are there official archival versions of the complete dataset (i.e., including the external resources as they existed at the time the dataset was created); c) are there any restrictions (e.g., licenses, fees) associated with any of the external resources that might apply to a future user? Please provide descriptions of all external resources and any restrictions associated with them, as well as links or other access points, as appropriate.** The creation of the dataset depends on the Jericho framework https://github.com/microsoft/jericho but the archival versions themselves do not have any dependencies.

**Does the dataset contain data that might be considered confidentiality, data that includes the content of individuals non-public communications)? If so, please provide a description.** No, all data is part of games that are already public.

**Does the dataset contain data that, if viewed directly, might be offensive, insulting, threatening, or might otherwise cause anxiety? If so, please describe why.** The data is collected from games containing situations of non-normative language usage—describing situations that fictional characters may engage in that are potentially inappropriate, and on occasion impossible, for the real world such as running a troll through with a sword. Instances of such scenarios are mitigated by careful curation of the games that the data is collected from. The original Jericho framework [Hausknecht et al., 2020]—further verified by us in this work—uses a curated set of games found not to contain extreme examples of non-normative language usage. This is based on manual vetting and (existing) crowd-sourced reviews on the popular interactive narrative forum IFDB https://ifdb.org/.

## B.3 Collection

**How was the data associated with each instance acquired? Was the data directly observable (e.g., raw text, movie ratings), reported by subjects (e.g., survey responses), or indirectly inferred/derived from other data (e.g., part-of-speech tags, model-based guesses for age or language)? If data was reported by subjects or indirectly inferred/derived from other data, was the data validated/verified? If so, please describe how** We build off the popular text game simulator Jericho [Hausknecht et al., 2020], we have constructed a dataset dubbed JerichoWorld that maps text game state observations to both the underlying ground truth knowledge graph representations of the game and the set of contextually relevant actions that can be performed in that state.

**What mechanisms or procedures were used to collect the data (e.g., hardware apparatus or sensor, manual human curation, software program, software API)? How were these mechanisms or procedures validated?** To collect the $\langle s_t, a_t, s_{t+1}, r_{t+1} \rangle$ tuples we implement a basic agent that explores the game along a trajectory corresponding to a *game walkthrough*. Game walkthroughs are texts describing the solutions to games, generally retrieved from the internet, but already part of the Jericho framework. Walkthroughs, however, only present one possible solution to a game and solve all the core puzzles required to complete a game with the maximum possible score. To achieve greater coverage of the game's state space, our data collection agent stops off to explore by executing random valid actions for $n$ steps before resetting to the walkthrough.

**If the dataset is a sample from a larger set, what was the sampling strategy (e.g., deterministic, probabilistic with specific sampling probabilities)?** Randomly sampled actions are based on a random seed in Python's random package https://docs.python.org/3/library/random.html. We provide a seed and the specific package version.

**Who was involved in the data collection process (e.g., students, crowdworkers, contractors) and how were they compensated (e.g., how much were crowdworkers paid)?** Only the authors were involved, building on the contributions of the Jericho developers.

**Over what timeframe was the data collected? Does this timeframe match the creation time-frame of the data associated with the instances (e.g., recent crawl of old news articles)? If not, please describe the timeframe in which the data associated with the instances was created.** This dataset was developed over a period of 6 months, though the games used within date back to the 1970s.

**Were any ethical review processes conducted (e.g., by an institutional review board)? If so, please provide a description of these review processes, including the outcomes, as well as a link or other access point to any supporting documentation.** No human subjects were involved, no IRB process was undertaken.

### B.4 Preprocessing

**Was any preprocessing/cleaning/labeling of the data done (e.g., discretization or bucketing, tokenization, part-of-speech tagging, SIFT feature extraction, removal of instances, processing of missing values)? If so, please provide a description. If not, you may skip the remainder of the questions in this section.** Games were decompiled to extract attributes and ground truth knowledge graphs, the creation script is provided in the GitHub repo.

**Was the "raw" data saved in addition to the preprocessed/cleaned/labeled data (e.g., to support unanticipated future uses)? If so, please provide a link or other access point to the "raw" data.** No, raw binary game states were not saved and were converted to human readable text.

**Is the software used to preprocess/clean/label the instances available? If so, please provide a link or other access point.** Games were decompiled to extract attributes and ground truth knowledge graphs, the creation script will be provided in the GitHub repository.

### B.5 Uses

**Has the dataset been used for any tasks already? If so, please provide a description.** No.

**Is there a repository that links to any or all papers or systems that use the dataset? If so, please provide a link or other access point.** No.

**What (other) tasks could the dataset be used for?** There are many more tasks that can be framed for other challenges related to world modeling from this dataset. Some immediate examples: (1) offline reinforcement learning for game agents through imitation learning—predicting the sequence of actions that finish the game based on walkthroughs and reward information; (2) knowledge graph verbalization, a form of the standard data-to-text natural language processing task [Wiseman et al., 2017], in which we learn to generate text that is conditioned on a knowledge graph; and (3) description generation conditioned on the names of various objects, locations, and characters—with applications in long-form text generation domains such as automated storytelling [Martin et al., 2018, Fan et al., 2019b] and procedural generation of interactive narratives [Ammanabrolu et al., 2020a, Walton et al., 2020].

**Is there anything about the composition of the dataset or the way it was collected and prepro-cessed/cleaned/labeled that might impact future uses? For example, is there anything that a future user might need to know to avoid uses that could result in unfair treatment of individ-uals or groups (e.g., stereotyping, quality of service issues) or other undesirable harms (e.g., financial harms, legal risks) If so, please provide a description. Is there anything a future user could do to mitigate these undesirable harms?** Users should keep in mind that these come from games and can potentially describe non-normative situations.

**Are there tasks for which the dataset should not be used? If so, please provide a description** This dataset should not be used for tasks that involve direct physical interactions with humans, such as robotics.

### B.6 Distribution

**Will the dataset be distributed to third parties outside of the entity (e.g., company, institution, organization) on behalf of which the dataset was created? If so, please provide a description.** It is open-sourced.

**How will the dataset will be distributed (e.g., tarball on website, API, GitHub)? Does the dataset have a digital object identifier (DOI)?** The dataset will be open-sourced at https://github.com/JerichoWorld/JerichoWorld.

**When will the dataset be distributed?** It was first released in May 2021.

**Will the dataset be distributed under a copyright or other intellectual property (IP) license, and/or under applicable terms of use (ToU)? If so, please describe this license and/or ToU, and provide a link or other access point to, or otherwise reproduce, any relevant licensing terms or ToU, as well as any fees associated with these restrictions.** The dataset will be under an MIT license, this is indicated on the GitHub repository.

**Have any third parties imposed IP-based or other restrictions on the data associated with the instances? If so, please describe these restrictions, and provide a link or other access point to, or otherwise reproduce, any relevant licensing terms, as well as any fees associated with these restrictions.** No.

**Do any export controls or other regulatory restrictions apply to the dataset or to individual instances? If so, please describe these restrictions, and provide a link or other access point to, or otherwise reproduce, any supporting documentation.** No.

### B.7   Maintenance

**Who is supporting/hosting/maintaining the dataset?** Prithviraj Ammanabrolu will be responsible for maintenance.

**How can the owner/curator/manager of the dataset be contacted (e.g., email address)?** `raj.ammanabrolu@gatech.edu` or by filing an issue on the GitHub.

**Is there an erratum? If so, please provide a link or other access point** No.

**Will the dataset be updated (e.g., to correct labeling errors, add new instances, delete instances)? If so, please describe how often, by whom, and how updates will be communicated to users (e.g., mailing list, GitHub)?** Yes, more games will be added and corresponding data will be collected. Previous versions will be kept for backwards compatibility.

**If the dataset relates to people, are there applicable limits on the retention of the data associated with the instances (e.g., were individuals in question told that their data would be retained for a fixed period of time and then deleted)? If so, please describe these limits and explain how they will be enforced.** No.

**Will older versions of the dataset continue to be supported/hosted/maintained? If so, please describe how. If not, please describe how its obsolescence will be communicated to users.** Yes, versions will be archived on the GitHub repository.

**If others want to extend/augment/build on/contribute to the dataset, is there a mechanism for them to do so? If so, please provide a description. Will these contributions be validated/verified? If so, please describe how. If not, why not? Is there a process for communicating/distributing these contributions to other users? If so, please provide a description** They can fork and submit pull requests to the current repository if they wish to extend it—these will be validated in an open-source manner on GitHub via reviews of the extensions.

