# OpenReview forum: "Modeling Worlds in Text "
_NeurIPS.cc/2021/Track/Datasets_and_Benchmarks/Round1 — NeurIPS 2021 Datasets and Benchmarks Track (Round 1)_

### Official Review · Reviewer_buPB · 2021-06-22
**All in all an interesting dataset with potential for many tasks**

**Rating:** 7
**Confidence:** 4

**Strengths:**

The dataset makes it easier to access game states and a limited number of reactions from text adventure games compared to having a learning agent interact with the text game itself. It is thus likely to help research in the development of world-exploring agents that interact with their environment through language.
It is reasonably big to allow for state-of-the-art deep learning techniques and provides a challenging evaluation setting. This is also exemplified by three baselines for two tasks presented in the paper.
The dataset has the potential to be useful for many more tasks beyond the two tasks discussed in the paper.

**Weaknesses:**

The data are limited to what was seen during a particular walkthrough. The additional random exploration does provide more variability but evaluating an agent in a "real game" would allow the agent to experience a reaction to any action it might propose. So what is the advantage of this dataset compared to creating or extracting the KGs on the fly while playing the game / letting the agent "play the game"?
I suppose accessing a text-based dataset is a lot easier and probably faster than running a game engine all the time but the question why or in which scenarios JerichoWorld should be preferred over just using Jericho is not discussed enough in the paper.

Ground truth graphs based on the omniscient game engine are of limited usefulness. Shouldn't an agent rather be trained to construct a mental state of the world based on its observations instead of being encouraged to guess plausible hidden facts (like what kind of object is hidden in a locked chest)? An agent should probably learn that there is usually a reward when anything locked is finally unlocked but I don't see how it is useful to mix observed parts of the world and wild guesses at unobserved parts.
It would be a very useful addition to the dataset if there was an annotation telling me whether an agent can be expected to know about a certain KG fact or not. It would also be a more interesting evaluation, in my opinion, to separately judge a system's observation and reasoning capabilities.
Furthermore, I disagree with the example mentioned in the paper where "commonsense reasoning" should tell the model that there is probably a sword in the locked chest and not something entirely different (unless the paper failed to mention that some description of the environment tells me that the chest is hidden in the "cave of the magic sword" or so). In an ideal dataset, KG facts that cannot possible be inferred from the environment should be marked as such. If that is not possibly due to the data collection method, this should be discussed as a limitation of the dataset.

**Additional Feedback:**

It is a minor point but the paper could explain better why KGs are a suitable representation for the dataset beyond the fact that previous work found them to be a good match for building a mental state of the world while training agents to navigate text adventures. Especially, challenges in KG generation could be discussed because it is a major part of one of the two proposed benchmark tasks.
In your opinion, might the data format introduce any additional challenges that are not strictly necessary to build a formal representation of the world's state?

**Clarity:**

The paper is well written and, at all times, I could follow the authors' line of thoughts.

The formalization of the two proposed tasks is the only part to criticize here. The tuple notation is ok for describing the data in the dataset generally but for the actual task descriptions the input should be separated more from the target data. The different between what is given and what should be predicted should be clearer.

**Correctness:**

The data collection method seems reasonable except what I mentioned in the 'Weaknesses' section about KG facts that cannot possible be inferred from the input. It is not necessarily incorrect to keep those KG facts but this property of the benchmark should be discussed more.

The seq2seq baseline for KG state prediction was trained to generate the linear serialization of actually graph-structured data. This is not a very common approach to knowledge graph extraction from text. I do not demand a more sophisticated baseline, e.g., based on relation extraction or graph rewriting, but it would probably strengthen the paper to mention that graphs are often enough treated as sequences, e.g., for graph-to-text generation [1], and that even generating graph serializations has been done successfully before [2].
As large parts of the previous state KG are also probably left as-is for this task, I think that a copy mechanism [3] has the potential to greatly improve the seq2seq baseline.
Besides possibly improving the baseline performance, this would also make the comparison and discussion of the different baselines more interesting because it would help with one of the big disadvantages of the seq2seq model by giving it an extractive component.

[1] https://www.aclweb.org/anthology/D19-1428/
[2] https://www.aclweb.org/anthology/2020.emnlp-main.577/
[3] https://www.aclweb.org/anthology/D19-1318/

**Documentation:**

The documentation of the dataset is excellent. I could not find the code for data extraction but only the data in the GitHub though.
I think the baselines are sufficiently described to be reproduced.

**Ethics:**

I do not have ethical concerns about the dataset.

**Relation To Prior Work:**

Cf. the section on "Weaknesses". In my opinion, it should be discussed more what advantage using this dataset has compared to using the Jericho framework alone.
Otherwise, this work is well motivated by previous work.

**Summary And Contributions:**

The authors provide a new dataset scraped from walkthroughs of 27 text adventure games with additional limited random exploration.
In each step, the dataset provides a current state of the world represented as a set of knowledge graph triples and textual descriptions of the virtual environment, as well as items with their attributes.
The dataset is reasonably big and features different genres of text-based games, which makes it an interesting resource. The knowledge graphs and item attributes were directly extracted from the games, which means that they reflect the internal game state accurately.

(modified: 20 Jul 2021)
The rebuttal addresses most of my concerns and I am confident that a better explanation in the text will help with most misunderstandings. The promised reprocessing of the data, which will include a special field to distinguish KG facts that can be extracted from the text and those that have to be inferred, will add a lot of value to the dataset. I therefore raise my rating to 7.

---

### Official Review · Reviewer_ZQNj · 2021-07-02
**A dataset of knowledge-graph prediction based on text-based adventure games**

**Rating:** 7
**Confidence:** 4

**Strengths:**

This dataset relates to a relatively small domain in NLP, which does not have  lots of reaserchers and many datasets. Thus it is a very welcome addition that might contribute to strengthen this 'niche area'  (which is a very interesting area !).
Moreover, this dataset seems to have a potential to be used in research in the wider scope of knowledge graph prediction, and that might be useful too.

**Weaknesses:**

The baseline models are evaluated on a fixed held-out test set. This can be useful for benchmark comparison when newer models are tested on the dataset.  However, a cross-fold validation might be a more robust approach here,
as it is not quite clear whether the held-out set (which is from other games than the training set) is similar (in relevant aspects) to the training set, or maybe very different.

**Additional Feedback:**

Some typos found in the manuscript:

1.
"bean search" on page 6, should be "beam search".

2.
Page 7.
"Based on this, we further note a trend in Table 3 that negative correlation between the as seen in Table 1."
Obviously something is missing in this sentence, before the word 'as'.
Also, I could not see a negative correlation in Table 3. It would be helpful if the authors explain how to read that in the table.

3.
Page 9.
"an comparison" -- should be 'a comparison'.

**Clarity:**

The paper is mostly clear.
There are several local issues that might require some clarification.

Section 3.1 mentions "input vocabulary" and "output vocabulary".
Most readers may not quite understand what those labels mean in the given context.
The paper states that input vocabulary "vocabulary that can be used to act in any of these games". Does it mean that those are the words that a game-engine would interpret if user uses them (and would igore any others)? (just out of curiosity - what about muti-word units?). Output vocabulary seems to be what a game engine may use in state descrptions (text output to user).
It may be helpful to provide more descriptions in that section, so the readers don't need to guess it.

It is not quite clear what "walkthroughs" are, a bit more description would be helpful, perhaps a small example.


**Correctness:**

The data set seems to be constrcuted in a sound way. Notably, this is quite an unusual dataset and few of its kind are available.


**Documentation:**

The dataset seems to be described in a sufficient manner (for a reasonable NLP practitioner to understand it).
Documentaion is provided in appendix B.
A URL to github is provided, and is functonal.
The description on github looks useful as it describes the data structure and fields.
Replication of the baseline results would not be trivial, but potentialy possible.
There are three systems (one of them from another publication). Description of the rule-based system is quite sketchy.
Implementation details are provided in Appendix A, including (hyper)parameter settings, etc.




**Ethics:**

The authors mention one particular concern - the mention or depiction of violent or 'unacceptable' scenes/actions in games, and this is a valid concern. As stated by the authors, all textual data was curated to prevent such language from occurring in the dataset.
I do not see any other ethical concerns.

**Relation To Prior Work:**

The paper describes recent prior work in detail (section 2).
Might want to add this to the list:
https://aclanthology.org/P18-1077/

**Summary And Contributions:**

This paper describes a dataset that is related to modeling in  interactive narratives (text-based adventure games). This is related to modeling situated agents that would  be able to 'intelligently' describe their 'world' and plan actions in it, which is a an interesting and complex problem area in AI and NLP.
The dataset provides mappings between NL observations (statements) and a knowledge-graph of states and objects (derived from the text game-engine simulator Jericho). The data is collected frrom 27 games for the training set (24K instances), and 9 additional games that serve as a testing set (7.8K instances).

Given that this area of research is not a standard type of tasks in NLP, the paper also defines two specific tasks for this dataset.
(1) Given a textual observation, predict the underlying knowledge graph of the world.
(2) Given a textual observation, predict the set of actions that are contextually relevant.

The paper further provides three baseine models for performing the first task (knowledge-graph prediction) - a rule-based model, a question-answering (QA) neural model, and a seq2seq model. The QA model outperforms the other two. For the second task (action prediction), only a Seq2Seq model is developed. As the authors note, there is much room for performance improvement on both tasks.

---

### Official Review · Reviewer_SN2Z · 2021-07-05
**The paper proposes a dataset for the development of knowledge-graph based world models of interactive narratives. Aside from some inconsistencies (see weakness section), the dataset is a valuable contriubution and should spark new research.**

**Rating:** 7
**Confidence:** 3

**Strengths:**

The specifics of the dataset are well-explained, and the importance of tasks associated with this dataset are clear. The tasks related to interactive narratives originates from AIIDE/INT communities and it is valuable to see these tasks formalized as part of the Neurips benchmarking dataset by developing a dataset which other researchers can use.




**Weaknesses:**


Although it is stated that Seq2Seq can perform common-sense inference by "filling in information missing from the [...] inputs," the authors have not provided an empirical example of such an instance as they had for the previous over-extraction case.

The analysis of the baseline Seq2Seq model on Valid Action Prediction appears to have inconsistencies.
(1) An important statement in the analysis of Valid Action Prediction is missing words: "Based on this, we further note a trend in Table 3 that negative correlation between the as seen in Table 1."
(2) It is stated that "the more the average number of gold standard valid actions per instance in a game, the more predicted actions match," which contradicts the statement about the negative correlation between these metrics. Table 3 shows that for games with more valid actions, the baseline model's predictions have less exact matches, which is not counter intuitive as the authors have suggested.
As a quick example, deephome has the lowest EM and F1, but has the second-highest average number of gold-standard valid actions.

Grammatical error in the middle of page 4, the middle of page 5, the bottom of page 8.


**Additional Feedback:**

N/A


**Clarity:**

The paper is generally well written with the exception of a few typos and grammar issues (see above)



**Correctness:**

The exposition was consistent with the exception of the analysis for Valid Action Prediction, which should be revised (see above)

**Documentation:**

Documentation is provided in the repository, and the formalism for the dataset's representation is provided in the main text. The documentation seems to be sufficient for researchers to use the dataset for their own research.



**Relation To Prior Work:**

Seems ok.

**Summary And Contributions:**

** Updated review 7/19 **

I have read the discussion and am satisified with the authors responses. I have increased my score to reflect this.

***

The authors provide a dataset and benchmarks for textual world modeling.
The dataset includes knowledge graphs that reflect the story world's state and a set of permissible actions for each observation.
Three baseline models are provided for the task of knowledge graph prediction, and 1 baseline model is provided for valid action prediction.

---

### Decision · Program_Chairs · 2021-07-26

**Decision:**

Accept

**Comment:**

All three reviewers agree that the paper provides an interesting dataset for textual world modeling along with the experiments for two specific tasks (i.e., knowledge graph prediction and valid action prediction). This resource could be helpful to make progress towards the development of agents that interact with their environment using natural language. Acceptance as a poster is recommended.